# Machine Learning-Based Alzheimer’s Disease Stage Diagnosis Utilizing Blood Gene Expression and Clinical Data: A Comparative Investigation

**DOI:** 10.3390/diagnostics15020211

**Published:** 2025-01-17

**Authors:** Manash Sarma, Subarna Chatterjee

**Affiliations:** Department of Computer Science and Engineering, Faculty of Engineering and Technology, Technology Campus (Peenya Campus), Ramaiah University of Applied Sciences, Bengaluru 560058, India

**Keywords:** Alzheimer’s disease stage diagnosis, blood gene expression, data imbalance, multiclassification, F1-score, AD risk gene, SHAP and LIME

## Abstract

**Background/Objectives:** This study presents a comparative analysis of the multistage diagnosis of Alzheimer’s disease (AD), including mild cognitive impairment (MCI), utilizing two distinct types of biomarkers: blood gene expression and clinical biomarker samples. Both of these samples, obtained from participants in the Alzheimer’s Disease Neuroimaging Initiative (ADNI), were independently analyzed utilizing machine learning (ML)-based multiclassifiers. This study applied novel machine learning-based data augmentation techniques to gene expression profile data that are high-dimensional, low-sample-size (HDLSS) and inherently highly imbalanced. The investigation obtained the highest multiclassification performance to date in the multistage diagnosis of Alzheimer’s disease utilizing the blood gene expression profiles of Alzheimer’s Disease Neuroimaging Initiative (ADNI) participants. Based on the performance results obtained, and other factors such as early prediction capabilities, this study compares the efficacies of the two types of biomarkers for multistage diagnosis. This study presents the sole investigation in which multiclassification-based AD stage diagnosis was conducted utilizing blood gene expression data. We obtained the best multiclassification result in both modalities of the ADNI data in terms of F1-score and were able to identify new genetic biomarkers. **Methods:** The combination of the XGBoost and SFBS (Sequential Floating Backward Selection) methods was used to select the features. We were able to select the 95 most effective gene probe sets out of 49,386. For the clinical study data, eight of the most effective biomarkers were selected using SFBS. A deep learning (DL) classifier was used to identify the stages—cognitive normal (CN), mild cognitive impairment (MCI), and Alzheimer’s disease (AD)/dementia. DL, support vector machine (SVM), gradient boosting (GB), and random forest (RF) classifiers were used for the AD stage detection from gene expression profile data. Because of the high data imbalance in genomic data, borderline oversampling/data augmentation was applied in the model training and original samples for validation. **Results:** Utilizing clinical data, the highest ROC AUC scores attained were 0.989, 0.927, and 0.907 for the identification of the CN, MCI, and dementia stages, respectively. The highest F1 scores achieved were 0.971, 0.939, and 0.886. Employing gene expression data, we obtained ROC AUC scores of 0.763, 0.761, and 0.706 for the CN, MCI, and dementia stages, respectively, and F1 scores of 0.71, 0.77, and 0.53 for CN, MCI, and dementia, respectively. **Conclusions:** This represents the best outcome to date for AD stage diagnosis from ADNI blood gene expression profile data utilizing multiclassification techniques. The results indicated that our multiclassification model effectively manages the imbalanced data of a high-dimension, low-sample-size (HDLSS) nature to identify samples of the minority class. MAPK14, PLG, FZD2, FXYD6, and TEP1 are among the novel genes identified as being associated with AD risk.

## 1. Introduction

An individual diagnosed with AD develops amyloid plaque, tau, and neurofibrillary tangles in the brain. There is a loss of connectivity between the neurons in the brain. The hippocampus is the likely place where the problem seems to originate. As the neurons degenerate, other brain parts are also subsequently affected. This leads to cognitive impairments such as short-term memory loss in the initial stages. Subsequently, the progressive deterioration of short-term memory occurs. This is followed by a decline in other cognitive faculties and the emergence of behavioral issues. Three broad stages are generally defined for the progression of this disease: CN, MCI, and AD. “Mild Cognitive Impairment (MCI) is attractive because it represents a transitional state between normal aging and dementia” [1]. Approximately 50–75% of individuals aged 65 years or above are generally susceptible to AD. As life expectancy increases, the prevalence of AD is also rising globally.

No major cure for AD has been established to date [2]. In this situation, the early diagnosis of the disease stage appears to be the only viable alternative. This would help minimize progressive deterioration through preventive actions. This creates critical research questions: What are the factors causing AD? Is it possible to utilize these factors for the early detection of the disease? What are these early biomarkers of AD, and the challenges associated with them? What is the potential of these early biomarkers for accurate AD stage prediction? Our aim was to find answers to these questions in this study.

While PIB, MRI, MMSE, CSF, and some other biomarker data are efficiently used in clinical AD diagnosis, omics data such as genomic biomarkers can help identify the AD risk in an individual much earlier than clinical symptoms of AD appear. In Figure 1, the “Systems Biology Approach” block has two parts—“molecular networks/pathways” and “healthy vs. disordered brains”. These two parts are surrounded by blocks representing multi-omics data (genomic, transcriptomic, epigenetic, environmental, metabolic, etc.). Multi-omics data such as the genome and transcriptome potentially impact the “molecular networks/pathways” that can lead to healthy or disordered brains, the second part of the block. These multi-omics data are factors that cause AD. A study of these factors can help in detecting the risk of AD much earlier [3]. We considered these datasets as early biomarkers. Other multi-omics data such as connectome, structural, and functional brain imaging reflect the brain structure, connectivity, and functionality of healthy and disordered brains, the second part of the “Systems Biology Approach”. This second part is the impact of the first part (molecular networks/pathways). The brain structure, connectivity, functionality, etc., ultimately impact the behavior of an individual. We used these data in AD diagnosis and considered them as biomarkers at the clinical stage [3].

Genomic factors play a major role in 80% of AD cases [5]. Genome Wide Association Studies (GWAS) can discover some AD candidate genes. However, GWAS have majorly failed to reliably produce AD candidate genes with reliability. Thousands of genes are considered as potential AD risk factors [6,7]. However, GWAS only discover genes that are associated with some phenotypes and fail to address the genes’ functionality causing AD [8]. Gene expression provides the opportunity for biochemical pathway analysis, regulatory mechanisms, and cellular functions to find the key AD and MCI genes. Some research utilized gene expression values from brain tissues from biopsy- or autopsy-based samples [9,10]. However, various difficulties are involved with such autopsy samples for analysis. Alongside brain dynamics, changes are also expressed in blood and a large portion of gene expression in the body is also found in PBMCs (Peripheral Blood Mononuclear Cells) [11]. Amyloid precursor protein expression and oxidative damage in the RNA and DNA of AD brain tissue are also reflected in peripheral blood [12]. So, blood gene expression is gaining recognition as a viable approach for diagnosing Alzheimer’s disease and mild cognitive impairment [13].

The clinical biomarker AD dataset encompasses measurements of brain structural integrity via MRI regions of interest (ROI), primary cognitive evaluations, the measurement of cell metabolism with FDG PET ROI averages, amyloid-beta load quantification through AV45 PET ROI averages, biomarkers for measuring tau load in the brain, axon-related microstructural parameters assessed with DTI ROI, CSF biomarkers for measuring tau and amyloid levels in cerebrospinal fluid, and additional data such as demographic information and APOE status including the number of APOE4 alleles. The APOE4 allele increases the risk of late-onset Alzheimer’s disease [5]. Clinical biomarker data and demographic variables such as age, gender, ethnicity, education, and marital status are frequently utilized in the clinical diagnosis of AD. With the increasing accessibility of biomarker datasets in the public domain, machine learning (ML) serves as a crucial aid in disease diagnosis and staging. ML is now widely used in disease diagnosis with the increasing availability of AD datasets.

Figure 2 illustrates the research steps involved in the comparative investigation of clinical and preclinical aspects of diagnosis. A literature review is conducted to assess the potential of clinical and preclinical biomarkers in Alzheimer’s disease (AD) prediction, the inherent challenges associated with each biomarker type, and the research gaps in addressing these challenges. Based on these findings, methodologies are implemented to elucidate the potential of each individual biomarker type for the stage diagnosis of AD. We design strategies to enhance performance in AD stage identification, evaluate the outcomes, and subsequently draw conclusions.

Because of the aforementioned limitation of GWAS, our study does not employ SNP- based genomic biomarkers as early biomarkers of AD diagnosis. Instead, this investigation utilizes blood gene expression profiles of ADNI participants as early biomarkers indicative of potential AD risk factors. Participants providing gene expression profile samples are included among those contributing clinical study samples. This approach would enhance the opportunity to comprehend and evaluate the inherent challenges and advantages of each type of biomarker, as well as applying appropriate machine learning techniques for predicting stages of Alzheimer’s disease. Additionally, the prediction performances of both approaches are compared and analyzed. Through this approach, it is anticipated that the previously mentioned research questions will be resolved.

### 1.1. Related Work

A substantial amount of research has already been conducted on AD diagnosis using ML techniques from diverse biomarker data including demographic and socioeconomic, to clinical and cognitive, imagery, genetic, and linguistic performance indicators. Numerous research articles have been published on this subject. To achieve the intended outcome of our study, with a focus on the clinical and preclinical aspects of diagnosis, we restricted our AD literature review to clinical and gene expression profile-based AD diagnosis. Among these, six research papers pertain to machine learning-based AD diagnosis from the imagery category of MRI and PET scan images. Three studies pertain to diagnosis based on gene expression profiles. One publication addresses multimodal analysis of gene expression, clinical, and SNP data. Another work focuses on diagnosis based on clinical, demographic, and non-imagery biomarker data. Table 1 presents a selection of research papers on AD diagnosis using machine learning methods, followed by a detailed exposition.

Westman et al. conducted research with CSF and a baseline MRI biomarker combination to enhance accuracy [14] compared with when each biomarker was used individually. Their dataset comprised 96 samples from 273 CN and 96 AD patients. Their proposed classification achieved an accuracy of 91.8% utilizing a combination of CSF and MRI. Veeramuthu et al. employed PET brain images to classify subjects with AD and CN. Initially, they conducted feature extraction with spatial normalization and noise filters [20], and then applied the Fisher discriminant ratio for feature extraction to obtain the region of interest (ROI). The associative rule (AR) mining algorithm was used for training, resulting in 91.33% accuracy, 100% specificity, and 82.67% sensitivity. However, their approach did not address the handling of missing data, the management of data with an imbalanced class, or the validation process. Fulton et al. designed a gradient-boosted machine (GBM) to predict AD from cognitive and sociodemographic data with 91.3% accuracy. Their deep learning and convolutional neural network-based ResNet-50 framework achieved a multiclassification accuracy of 98.99% from MRI [15]. However, the findings cannot be generalized due to the limited sample size of merely 416 persons in the study. Lella et al. developed an ML framework for the classification of AD and the analysis of feature significance using DICOM images obtained from the ADNI database [21]. Artificial neural network (ANN), support vector machine (SVM), and random forest (RF) techniques were applied to classify the information content of the communicability metric of the image samples collected from AD and CN natives. By applying image processing techniques, they generated a connectivity matrix that represented the structural complexity of the brain network of each subject. Their ANN model attained an AUC score of 83% and an accuracy of 75% in the classification task. Despite a competitive performance, the study omitted the MCI stage of classification. Sarma et al. [22] applied different machine learning techniques to predict AD stages (CN, MCI, and AD) and achieved the optimal F1 scores of 89% for CN, 84% for MCI, and 80% for AD stage identification using deep learning from ADNI baseline clinical trial samples from 2000 participants. They utilized Sequential Floating Backward Selection (SFBS) and correlation matrix techniques to reduce dimensionality from 113 biomarkers to 8 biomarkers. Nonetheless, DL techniques are typically sensitive to the nature of training data. They are also sensitive to random initialization, which is often overlooked and has not been addressed.

In their study, Bae et al. [23] utilized T1-weighted MRI images of the medial temporal lobe to detect AD. They employed Inception-v4, a pretrained 2D image classification CNN model. The research utilized two datasets: one from ADNI and another from Seoul National University Bundang Hospital (SNUBH). The fine-tuned model comprised 156 AD patients and 156 CN controls from each dataset while testing the final model was conducted with 39 AD patients and 39 CN from each dataset. Five model instances were generated from each dataset utilizing 5-fold cross-validation. The final outcomes of the test were determined by calculating the mean ensemble values of the average probabilities derived from models generated through cross-validation. The ADNI-trained model achieved an AUC score of 0.94 using ADNI test data while the SNUBH-trained model attained an AUC score of 0.91 using SNUBH test data. The AUC ratings for the ADNI-trained–SNUBH test and SNUBH-trained–ADNI test data were 0.88 and 0.89, respectively. Fathi et al. [19] selected six of the most effective individual CNN-based classifiers to combine and construct an ensemble model for classifying AD stages. They achieved accuracy rates of 98.57, 96.37, 94.22, 99.83, 93.88, and 93.92 for NC/AD, NC/EMCI, EMCI/LMCI, and LMCI/AD, four-way, and three-way classification groups, respectively, from the ADNI MRI dataset. The ensemble technique significantly enhanced performance compared to individual model performance; however, when tested with a local MRI dataset, the performance was suboptimal. Furthermore, they employed accuracy as a performance metric for multiclass classification, which is not recommended.

Regarding the gene expression data modality, datasets from NCBI and ADNI have been predominantly utilized in research on AD diagnosis employing statistical methods, machine learning algorithms, and deep neural networks. Substantial research on Alzheimer’s disease utilizing genome expression data has been conducted employing machine learning techniques. Huang et al. [24] developed an SVM-based method to classify AD genes from gene network data of the human brain and gene expression in the whole genome. The candidate genes of AD were classified with an accuracy and ROC of 84.56% and 94%, respectively. This methodology provides a complementary approach for the spectrum of AD-associated genes identified from more than 20,000 genes on a genome-wide scale. Lee et al. in their study identified AD-related genes from DEGs (Differentially Expressed Genes), the TF (Transcription Factor) database, gene connectivity network data, and CFG (Convergent Functional Genomics) from blood gene expression data [9]. The AD-related gene expression data were utilized to construct ML models from logistic regression (LR), L1-regularized LR (L1-LR), SVM, RF, and DNN classifiers. The best average values of the AUC (area under the curve) obtained were 0.657 for ADNI, 0.874 for ANMI, and 0.804 for the ANM2 dataset employing five-fold cross-validation for each dataset. These findings suggest that gene expression data derived from blood samples demonstrate potential utility in predicting AD stages. Nevertheless, the significant data imbalance in the ADNI dataset, where the minority dementia sample size is very small compared with the other two categories, results in a suboptimal AUC score in the case of ADNI. Additionally, multiclassification is avoided here, even though the datasets have three labels, including the MCI stage. Park et al. in their research [16] proposed a DL-based model capable of classifying AD by integrating large-scale DNA methylation and gene expression data to construct multi-omics datasets. They attained an accuracy of 0.823, surpassing that of a single omics dataset. However, they did not use the multi-omics dataset from the same sample cohort. AlMansoori, M.E. et al. [18] developed a multimodal machine learning method to differentiate between CN and MCI/AD cases. The study demonstrated an AUC performance of 0.95 when combining gene expression profiles, clinical data, and SNP data using Mutual Information (MI) feature selection from 623 ADNI participants in a binary classification approach. However, when utilizing gene expression data independently, the AUC performance decreased to 0.65. Furthermore, integrating genetic data with clinical data could potentially enhance performance. Nevertheless, it remains to be elucidated how efficacious it is to combine preclinical data with clinical data, considering that preclinical data are typically utilized to predict disease significantly earlier than the onset of clinical symptoms.

### 1.2. Research Gap Analysis

Primary limitations in the earlier research were the lack of sufficient and authentic data samples and lower accuracy achieved [22]. Nevertheless, a constraint in the majority of previous research utilizing gene expression and clinical data is the omission of the mild cognitive impairment (MCI) stage, the intermediate stage of Alzheimer’s Disease (AD) in diagnosis, despite obtaining impressive scores with binary classification. This omission of the MCI stage, which constitutes a crucial phase of the disease progression, represents a significant aspect of research that has been inadequately addressed [1]. Recent research incorporating the MCI stage has yielded notable results in the binary classification of disease states, specifically AD versus CN, AD versus MCI, and MCI versus CN. Although employing binary classification may have yielded improved results, this approach necessitates the use of three distinct models and potentially leads to misclassification when evaluated with previously unseen samples encompassing all three AD stages. MCI stage is a preclinical stage that can significantly indicate the progression of AD.

Subsequently, the performance output of some of the earlier research demonstrates promise with an AUC score around 80% when balanced GEO datasets are used for model training/testing. However, results are not promising when the training and test dataset is ADNI. In the research conducted by Lee H. et al. [9] with ADNI data, the AUC for binary classification of CN against AD was 0.657 during internal validation (using ADNI data for training and testing/validation). The score is significantly lower when compared with other outcomes. One potential cause for the suboptimal performance is that ADNI gene expression data exhibit HDLSS characteristics [25] and are also imbalanced. The dataset comprises 49,386 gene probes/gene transcripts (or features), contains only 744 samples, and exhibits an imbalance, as illustrated in Table 2. The authors employed novel feature selection and modeling approaches; however, they did not adequately address the impact of low sample size and inherent data imbalance on the model’s learning capacity.

While certain effective gene selection techniques have been employed, it would have been more comprehensive to consider the entire list of genes as input for selection. Regarding limited sample size and data imbalance, certain researchers have addressed this issue by augmenting the sample size through the integration of comparable gene expression datasets, such as ANM1 and ANM2, which are affiliated with NCBI. This approach is not feasible for ADNI gene expression profile data, as the samples are collected utilizing different technology from that of NCBI. Consequently, the diagnostic performance of models derived from ADNI gene expression profiles is substantially limited [10]. Although inherent complexity is associated with gene expression data and cannot be circumvented, enhanced feature selection and classification techniques, in conjunction with data augmentation, would have significantly improved classification performance.

Some researchers [18] integrated preclinical stage biomarkers with clinical biomarkers and were able to produce exceptionally high performance. While the integration of clinical biomarkers such as CSF and MRI and similarly the integration of preclinical biomarkers such as gene and epigenetic expression are logical, the integration of clinical biomarkers with preclinical biomarkers may not be efficacious in practical diagnosis. This is primarily attributable to the fact that preclinical biomarkers are predominantly utilized to identify the risk of Alzheimer’s disease well before the onset of clinical symptoms.

Furthermore, researchers did not employ appropriate performance metrics for AD classification measurement from imbalanced data, which is very common in real data. They used accuracy and AUC as performance measurements and omitted F1-score. F1-score provides a more accurate assessment of model performance across all types of test data, particularly for imbalanced data such as ADNI blood gene expression data.

### 1.3. Key Contributions

This study presents four primary contributions. First, we conducted stage diagnosis utilizing two distinct data modalities of ADNI participants: blood gene expression profile, a biomarker at the preclinical stage, and clinical study data, a biomarker at the clinical stage. Given that both data modalities originate from the same ADNI source, with participants contributing gene expression samples belonging to the cohort contributing clinical study samples, we have a pragmatic comparison of the inherent challenges of both sets of biomarker data and their efficacies in multistage AD prediction. As illustrated in Figure 3, we developed two separate models and subsequently compared their performances and discussed the inherent challenges and efficacies of each type of biomarker in AD stage prediction.

Second, we achieved a multiclassification-based ROC performance score on ADNI genomic data, which significantly surpasses the AUC score of 0.657 and 0.65 obtained from binary classification with internal validation in two previous studies [9,18], as discussed in Section 1.1. Given the low efficacy in Alzheimer’s disease (AD) identification from ADNI blood gene expression data in prior studies, we opted for a unique feature selection and a novel data augmentation approach. Third, this study represents the first instance in which an F1-score-based performance analysis and multiclassification were implemented in the context of blood gene expression-based AD multistage assessment whereas prior research was limited to AUC-based performance analysis. Our research demonstrates the ability to achieve the highest reported F1 score and AUC score from non-imagery clinical data in Alzheimer’s disease diagnosis using multiclassification.

Fourth, this study uncovered novel genes that may be potentially implicated in the etiology of or confer protection against AD. The analysis of the second modality of the ADNI clinical dataset revealed the most effective biomarkers for diagnosis, achieving the highest known F1-score and ROC AUC for multiclassification in AD stage diagnosis. The results of both models developed utilizing the two modalities of ADNI subjects were analyzed.

## 2. Material and Methods

### 2.1. Dataset Selection and Exploration

In our investigation, we focused on two modalities of ADNI data: blood gene expression profile and clinical study datasets available on the Laboratory of Neuro Imaging (LONI) website. The 744 ADNI participants contributing to gene expression samples constitute a subset of the 2400 participants contributing to clinical study samples. The gene expression profiling data are derived from the blood samples of 744 ADNI participants. The samples were collected in different ADNI phases, and the Affymetrix Human Genome (HG) U219 Array was used for expression profiling. The samples were passed through different QC processes and ultimately samples from 744 participants were chosen for further analysis. Each sample from these 744 participants comprises a total of 49,386 probe sets. The acquisition of stage diagnosis information for the participants presented challenges and was derived from ADNI clinical/study data by utilizing collection phase/date and patient ID information common to both study/clinical and gene expression profile datasets. The participants in the genome expression profiling constitute a subset of the study/clinical data participants.

ADNIMERGE is the clinical study dataset utilized as the additional data modality in this research. The ADNIMERGE data were collected in multiple phases and comprised baseline and periodic samples. In this study, baseline clinical samples were collected from 2400 participants. The collected clinical dataset consisted of total 892 samples from the CN class, 1821 samples from the MCI class, and 413 samples from the AD or Dementia class.

The blood gene expression dataset, comprising 744 participants, is detailed in Table 2, which shows the sample distribution across three categories: CN, MCI, and AD.

### 2.2. Data Pre-Processing and Feature Selection

The ADNI clinical dataset exhibits relative simplicity in comparison to the gene expression profile, which represents the alternative modality. We utilized the SFBS algorithm for feature selection. Based on our examination of previous research on AD diagnosis, we selected 25 biomarkers from a total of 116. Subsequently, we applied the SFBS (Sequential Floating Backward Selection) technique, which identified 8 essential features from the initial 25, as illustrated in Figure 4 and Table 3.

Quality control procedures for gene expression data were implemented at the time of the public provision of the dataset, resulting in the selection of 744 samples. Subsequently, pre-processing and cleaning were directly applied to the gene expression dataset. Data normalization and scaling were performed using the ‘MinMax’ scaler. Following pre-processing, the dataset comprised 744 samples (rows) and 48,158 gene transcripts (features). This situation exemplifies the ‘curse of dimensionality’ due to the exceptionally high dimensionality [26]. Consequently, this leads to significant overfitting. Furthermore, the magnitude of approximately 50,000 features impacts computational efficiency. Previous research has employed feature selection techniques such as VAE (Variational Auto Encoder), LASSO, and RFE (recursive feature elimination) [9,17]. For autoencoder techniques, it is necessary to specify the number of features to be minimized, with the selected features represented in an encoded format.

This study employed a two-step process for feature selection. Initially, we calculated the feature score of each feature utilizing the XGBoost algorithm and subsequently ranked 48,158 genes according to these scores. Subsequently, we applied the SFBS algorithm to the top 300 gene transcripts. The SFBS algorithm automatically selected 95 features/transcript genes from the 300 transcripts, as illustrated in Figure 5 below.

As illustrated in Figure 6 below, the graph of the top 40 features indicates that the gene probe/transcript ‘SCGB1D4: 11737959_at’ is positioned at the bottom of the graph exhibiting the highest score. The SHAP summary plot reveals a hierarchy of influential features, with ‘SCGB1D4: 11737959_at’ demonstrating the highest impact. Subsequent features of significance include ‘ASXL3: 11737082_a_at’, ‘MAPK14: 11742204_a_at’, ‘TMEM30B: 11756750_a_at’, and ‘SQLE: 11717731_at’, followed by additional features. The score graph plot ranks these features within the top 25.

Recent research has utilized the SHAP explainability and feature selection tool, leveraging its visualization capabilities and other functionalities [27]. Employing an XGBoost algorithm-based model, we applied the SHAP method to determine the impact of each feature or gene transcript on the model output for three stages of Alzheimer’s disease (AD) patients: cognitively normal (CN), mild cognitive impairment (MCI), and dementia. We conducted SHAP analysis on the ADNI gene expression dataset. The SHAP analysis tool, when applied to these datasets, generates output figures that facilitate the understanding of the positive and negative impacts of gene expression values for each gene transcript. If a high gene expression value of a gene transcript produces a significant impact indicative of AD or MCI stage as output, the gene is considered an AD risk factor. Conversely, if it has a negative impact or if a high gene expression value of a gene transcript produces a significant impact indicative of CN stage as output, the gene is considered an AD suppressor gene.

In our investigation, the application of SHAP analysis to ADNI gene expression data resulted in the generation of Figure 7 and Figure 8. This facilitated the examination of the influence of various genes on AD stages. Subsequently, we identified genes associated with AD, as enumerated in Table 4 below.

### 2.3. AD Multistage Diagnosis Model Construction

Deep learning is the common classifier used for model construction for both data modalities. Given the availability of more than 2000 clinical samples, deep learning was deemed the appropriate methodology for developing a model from the clinical data. Deep learning is a better classifier technique than many other ML-based classifiers [28]. A prior study by the authors exhibited superior performance utilizing deep learning techniques with ADNI clinical data relative to other machine learning multi-classifiers [24]. Consequently, alternative machine learning classifiers have not been explored for analyzing clinical data. It is worth noting that prior researchers [9,16,18,23] have effectively applied SVM, RF, and boosting algorithms in blood gene expression studies, yielding encouraging outcomes. The gene expression dataset comprised 744 samples. There remains substantial scope for exploring additional machine learning algorithms. This study conducted experiments utilizing three additional machine learning-based classifiers: SVM, GBM, and RF.

#### 2.3.1. Multiclassification Model for Clinical Data

The deep learning model developed for ADNI clinical study data analysis exhibits a straightforward architecture. Unlike ADNI gene expression profiles, this dataset does not possess high-dimension, low-sample-size (HDLSS) characteristics. The clinical data comprised 2000 samples, which is sufficient for model construction. The available 2400 samples were partitioned into 5 sets of training and validation utilizing 5-fold cross-validation, with each set having a 20% allocation for validation [29,30]. Five deep learning model instances were trained from each set. The hyperparameters utilized during the training process are delineated in Table 5. The performance metrics of the five most effective deep learning model instances are presented in the subsequent section.

#### 2.3.2. Multiclassification Model for Gene Expression Data

The genomic dataset exhibits high-dimensional, low-sample-size (HDLSS) characteristics and is imbalanced in nature. The dataset comprises 384 samples of mild cognitive impairment (MCI), 280 of cognitively normal (CN), and only 116 of dementia, as indicated in Table 2. A significant imbalance exists between the MCI and dementia categories. The minority class, dementia, presents challenges in prediction due to its limited sample size, resulting in fewer learning opportunities compared to the majority samples. A common issue associated with imbalanced datasets is that most learning methods exhibit bias towards the majority class, leading to the inadequate modeling of minority samples [31]. Despite extensive research efforts to address imbalanced learning, numerous limitations persist [32]. To date, no research has been conducted on the classification of imbalanced gene expression data. The primary challenge in imbalanced classification stems from the insufficient samples in the minority category, which impedes the model’s ability to accurately learn the decision boundary [33].

As illustrated in Figure 9, there are primarily two approaches for developing various models from imbalanced data. Given the small overall sample size and its imbalanced characteristics, the data augmentation strategy was preferred over alternative methods such as weighted or ensemble modeling. The application of oversampling to obtain a more balanced dataset is an effective strategy. Minority sample data are replicated prior to model fitting. However, random oversampling increases the probability of overfitting occurrence as it creates exact copies of minority class samples [31]. This study employed oversampling techniques where additional minority samples are synthetically generated, which is a form of data augmentation. One of the most widely used methods is the ‘Synthetic Minority Oversampling Technique’, or SMOTE [34]. This research utilized a popular extension, ‘Borderline SMOTE’, where misclassified minority samples are selected for oversampling instead of indiscriminate oversampling. The samples along the borderline and those nearby tend to be misclassified more frequently than those distant from the borderline, making them more crucial for classification [35].

As illustrated in Figure 10 below, the data were randomly partitioned into 5 sets of training and testing samples utilizing 5-fold cross-validation. Oversampling was applied to the training dataset, but not to the validation dataset. This approach is necessary to ensure that model evaluation is conducted using a dataset that accurately represents the problem domain. Evaluating the model with a dataset containing deleted or synthesized examples would likely result in an overly optimistic performance estimation [33].

## 3. Performance Results

We utilized ROC AUC and F1-score to evaluate the model performance of multiclassification. Accuracy, a commonly employed performance metric, is recognized as an inadequate measure for model performance assessment with imbalanced data [33]. Consequently, we did not employ this metric. A range of metrics is utilized when learning from imbalanced data. The most prevalent among these are receiver operating characteristic (ROC) analysis and the area under the ROC curve [36,37]. Additional widely utilized metrics in learning from imbalanced data are derived from precision and recall. The F1-score measure equally weights precision and recall. Beyond F1-score and ROC AUC, researchers can utilize alternative performance metrics for imbalanced classification, such as G-Mean, Brier score, F2-score, and F0.5-score [38]. F1-score is appropriate when false negatives and false positives hold equal significance. However, when there is greater concern about false negatives, F2-score is more appropriate. Conversely, F0.5-score is preferred when false positives are of higher importance. Probabilistic metrics like the Brier score are tailored to assess the uncertainty in classifier predictions. In our study on stage identification, where all classes hold equal significance, we opted to use F1-score and ROC AUC as our evaluation criteria. Furthermore, the F1-score variant is frequently employed as a performance metric in learning from imbalanced data [33].

For both clinical data and genome expression data, we used the k-fold cross-validation technique. Cross-validation is a data resampling methodology utilized to evaluate the generalization capability of predictive models and to mitigate overfitting [29,30]. “In k-fold cross-validation, the available learning set is partitioned into k disjoint subsets of approximately equal size. Here, fold refers to the number of resulting subsets. This partitioning is performed by randomly sampling cases from the learning set without replacement. The model is trained using k–1 subsets, which, together, represent the training set. Subsequently, the model is applied to the remaining subset, which is denoted as the validation set, and the performance is measured. This procedure is repeated until each of the k subsets has served as validation set” [29].

An elevated k value would reduce the quantity of test samples while augmenting training samples. The genetic dataset comprised 744 samples, while the clinical study had 2400 samples. Given the relatively small amount of gene expression data, we set k = 5 for k-fold cross-validation for both gene expression and clinical data. The 5-fold cross-validation would retain 20% of the total samples for testing. With this, the testing sample count for gene expression data was 148, which is optimal, while the training sample count was 596. We established k = 5 for the clinical data-based model in cross-validation, consistent with the gene expression-based models to facilitate performance comparison. The number of training samples was augmented by the SMOTE-based minority oversampling method where the minority sample size was increased, as outlined in the preceding section.

For both datasets, we initially conducted multiple trials by temporarily modifying the random state in the k-fold cross-validation method to control randomness, ensure the reproducibility of the results, and optimize parameters. This facilitated the minimization of performance variations across diverse random states, resulting in consistent AUC and F1-score outcomes for each model instance generated from both biomarker categories, as illustrated in the subsequent tables.

Table 6 and Table 7 present the area under the curve (AUC) and F1-score results from the 5-fold cross-validation models of the deep learning (DL), support vector machine (SVM), gradient boosting machine (GBM), and random forest (RF) classifiers constructed from gene expression data. For each Synthetic Minority Oversampling Technique (SMOTE)-based model result, the associated result from a non-SMOTE-based model result is presented. The first row in the SMOTE-based section of the table presents the evaluation of each classifier constructed from the identical oversampled training set with the corresponding testing data from the first cross-validation. This corresponds to the first row of the non-SMOTE models constructed from the data of the first cross-validation. Subsequently, the second to fifth rows present the results of the complete 5-fold validation for both SMOTE- and non-SMOTE-based models.

The results indicate that, in the majority of instances, the SMOTE-based models exhibit superior performance compared to their non-SMOTE-based counterparts. Furthermore, it is evident that the stage diagnosis of CN and MCI stages demonstrates higher efficacy relative to AD/dementia stage identification. This can be attributed to the larger sample size in comparison to the minority AD samples, as elucidated in the preceding section. A larger sample size facilitates the enhanced learning of the specific features involved, while a low sample size results in reduced learning, thereby impacting the performance in identifying learned features. Nevertheless, the application of minority oversampling techniques results in an enhancement in AD/dementia stage identification performance. This improvement is evident when comparing the performance of each instance of SMOTE-based model with its non-SMOTE-based model counterpart. The results obtained demonstrate significantly superior outcomes compared to those of prior research conducted with ADNI gene expression data [9,18]. This establishes our superior feature selection and innovative data augmentation of minority samples for training.

Table 8 illustrates the performance metrics of the five clinical data-based deep learning model instances constructed using 5-fold cross-validation. The performance data, presenting F1-scores and ROC AUC values for the multiclassification of the deep learning model derived from clinical data, surpass those of gene expression-based models, as anticipated. This superiority is attributable to the substantially larger sample size, relatively straightforward data structure, and absence of challenges inherent in gene expression data, such as high-dimensionality, low-sample-size (HDLSS) characteristics, data imbalance, and multiple pathway involvement.

Figure 11 illustrates a visual comparative analysis of the best performing models constructed using gene expression data and clinical information. The evaluation metrics employed include the receiver operating characteristic (ROC) curve, the precision–recall (PR) curve, and the confusion matrix (CM). The graphical representations and confusion matrix outcomes provide compelling evidence for the superior multiclassification capabilities of a model derived from clinical data. The results indicate that both models demonstrate enhanced classification ability in identifying CN and MCI stages compared to the AD category where AD represents minority samples. To evaluate and compare the performance of both model types, we prioritized assessing the classification accuracy of the AD/dementia category, which represents the minority samples over other CN and MCI categories. As per the confusion matrix, out of 131 AD/dementia testing samples, the clinical data-driven model categorized 109 samples as AD/dementia and 22 as MCI, with no samples categorized as CN, whereas the blood gene expression model classified 8 samples as AD/dementia, 7 as MCI, and 3 as CN, from a total of 18 AD/dementia testing samples. The clinical data-based model exhibited superior classification performance compared to its gene-based counterpart due to multiple factors elucidated in other sections. However, the gene-based model effectively distinguished 15 combined AD/dementia and MCI samples from cognitive normal (CN) out of a total of 18 dementia samples.

## 4. Discussion and Conclusions

This investigation independently utilized gene expression data and clinical study data for the diagnosis of Alzheimer’s disease stages. The findings indicate that the model constructed using gene expression—an alternative data modality—demonstrates relatively low multiclassification performance for Alzheimer’s disease stage diagnosis compared to the model built on clinical data—the other data modality. Multiple factors contribute to this outcome. Primarily, the gene expression dataset, comprising 744 samples, is substantially smaller in comparison to the clinical study dataset, which contains 2000 samples. This considerable disparity in sample size inherently constrains the potential for comprehensive model training. Additionally, the dataset presents significantly more complexities compared to the clinical data, as previously elucidated. The initial two factors necessitate advanced techniques for identifying crucial features, subsequently followed by the multiclassification of the data, as we delineated in the preceding sections. The third reason pertains to the involvement of numerous pathways in genomic data. While gene expression is a major area of functional genomics, there are other domains such as proteomics, metabolomics, and epigenomics which contribute to variations in the type and quantity of protein production, leading to an individual’s phenotype, health status, and susceptibility to various diseases [39]. Figure 12 below elucidates this point. Although our research achieved the best outcome in multistage AD diagnosis utilizing ADNI gene expression data, the results do not match the performance of models derived from clinical data. Nevertheless, preclinical biomarkers such as genome expression, which is a prominent early indicator, have the potential to identify the AD stage significantly earlier than the manifestation of AD symptoms in an individual, whereas clinical biomarkers primarily capture symptomatic manifestations.

The varying performance scores of the models constructed from each data modality demonstrate promising results. Both outcomes demonstrate the effectiveness of our feature selection and model construction in each approach. Notably, an accurate diagnosis of AD stages can be achieved using only 8 clinical biomarkers, eliminating the need for an extensive panel of 116 clinical biomarkers. This finding has the potential to provide valuable support for physicians in laboratory settings. Through the implementation of a proficient feature selection methodology in conjunction with a novel data-augmentation technique for addressing imbalanced data, we identified 95 significant gene transcripts from a total of 49,386, thereby improving the ROC AUC score from 0.657 and 0.65, as reported in prior research on the binary classification of CN versus AD [9,18], to 0.763, 0.761, and 0.706 in the multiclassification of CN, MCI, and AD stages, respectively. An average gain of 8% in the AUC score is noteworthy, particularly considering the introduction of multiclassification in lieu of binary classification utilized in previous research. While none of the previous research studies on AD diagnosis with blood gene expression data considered F1-score for performance analysis, our research utilized F1-score in multiclassification and achieved the competitive scores of 0.71, 0.77, and 0.53 for CN, MCI, and AD, respectively, Our SHAP analysis of 95 gene transcripts identified a subset of genes potentially relevant to the etiology of Alzheimer’s disease (AD). This investigation did not elucidate a role for the known AD risk alleles of APP (amyloid precursor protein), PS1 (presenilin 1), PS2 (presenilin 2), and apolipoprotein E (ApoE). One potential explanation is that these gene alleles may remain dormant and incapable of eliciting AD, despite their recognized risk potential, if their expression levels do not reach a critical threshold. This observation supports the conclusion that Alzheimer’s disease does not have a singular genetic etiology. Rather, it may be influenced by the interaction of multiple genes in conjunction with other factors [39,40]. Figure 13 illustrates the LIME plot, which demonstrates various genes associated with dementia, including TEP1, SQLE, HLA-DQA1, TTC28, and ZNF835. Upon analyzing the contributing genes of a dementia test sample using LIME, we observed that the LIME results were consistent with these findings.

In conclusion, we successfully built a sophisticated multiclassification model for AD stage diagnosis including intermediate MCI stage utilizing two data modalities of clinical and preclinical biomarkers after taking care of multiple challenges. Both models independently demonstrated best performance in terms of F1-score and ROC AUC. Our investigation identified the factors contributing to the relatively suboptimal performance of models derived from gene expression data while emphasizing their contributions towards early prediction. Although gene expression data have inherent limitations in facilitating a very high accuracy, there is scope for improvement through integration with other preclinical genomic biomarkers such as epigenetic data. This aspect remains a subject for future research. Further investigation into the relationship between genomic biomarkers and clinical AD biomarkers reflecting AD symptoms presents an interesting avenue for potential future studies. Furthermore, this stage diagnosis approach utilizing gene expression and clinical biomarkers is expected to be more economically feasible due to its relatively lower cost, especially when juxtaposed with expensive alternatives such as PET, MRI, and other high-cost diagnostic techniques.

## Figures and Tables

**Figure 1 diagnostics-15-00211-f001:**
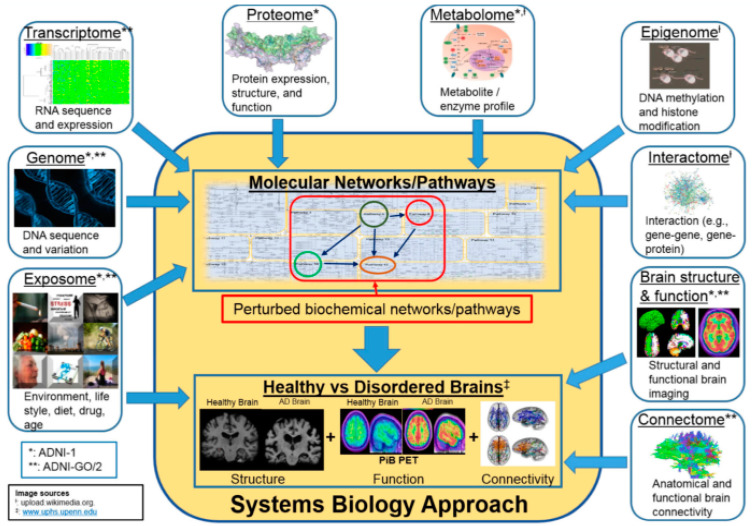
AD diagnosis—a system biology approach; source (s)—Wikimedia Com (^†^ http://upload.wikimedia.org/, ^‡^ http://www.uphs.upenn.edu/) and review paper [4]. Reprinted/adapted with permission from Ref. [4].

**Figure 2 diagnostics-15-00211-f002:**
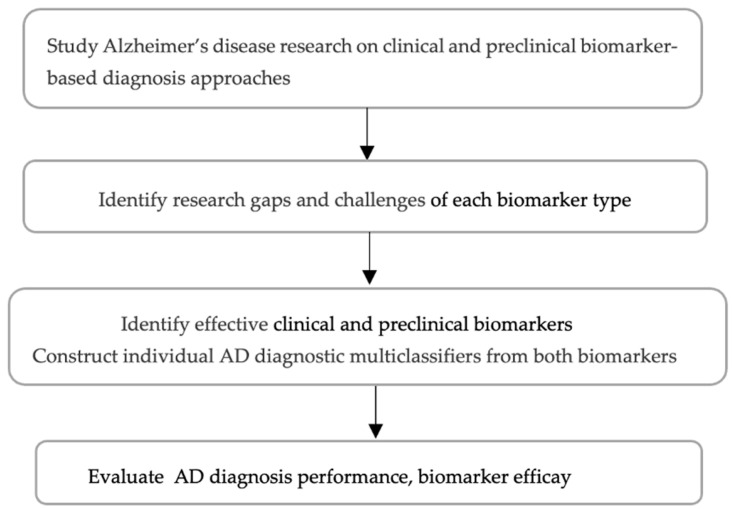
Research steps in comparative analysis of AD diagnosis.

**Figure 3 diagnostics-15-00211-f003:**
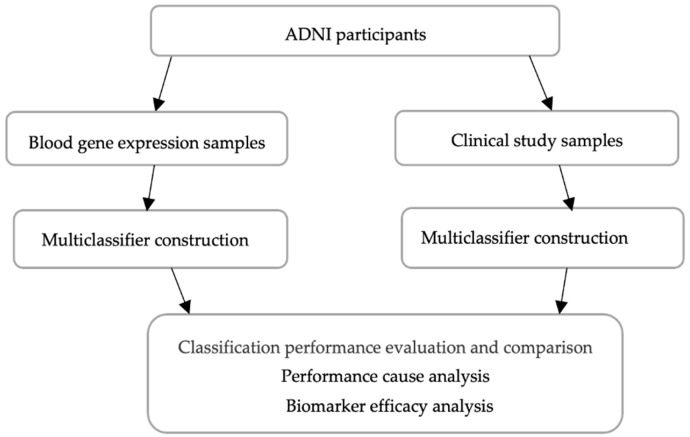
Comparative investigation of AD diagnosis at broad level.

**Figure 4 diagnostics-15-00211-f004:**
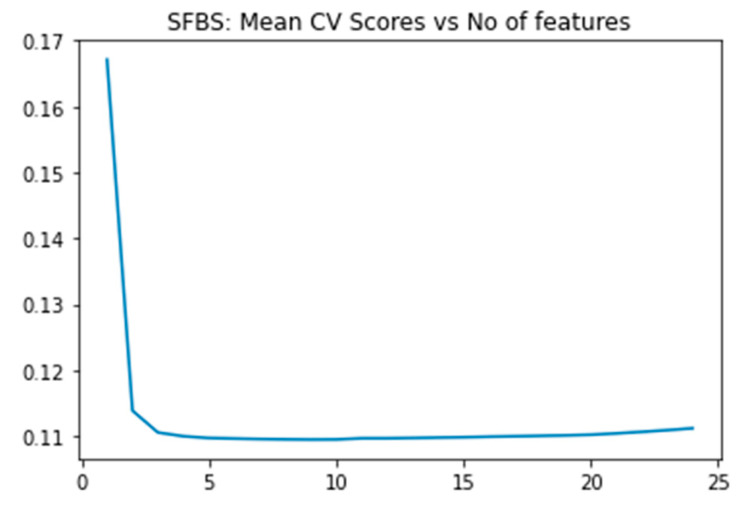
SFBS graph depicting the relationship between mean cross-validation scores and the number of features of selection of 8 clinical feature biomarkers.

**Figure 5 diagnostics-15-00211-f005:**
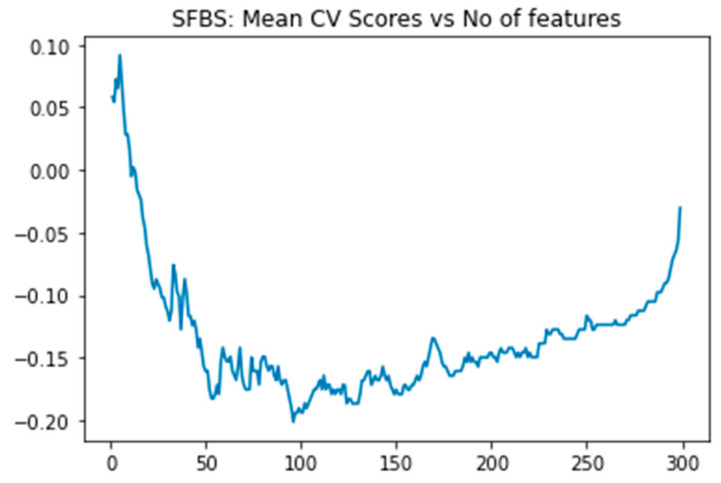
SFBS graph depicting the relationship between mean cross-validation scores and the number of features for selection of 95 gene features/gene probe sets.

**Figure 6 diagnostics-15-00211-f006:**
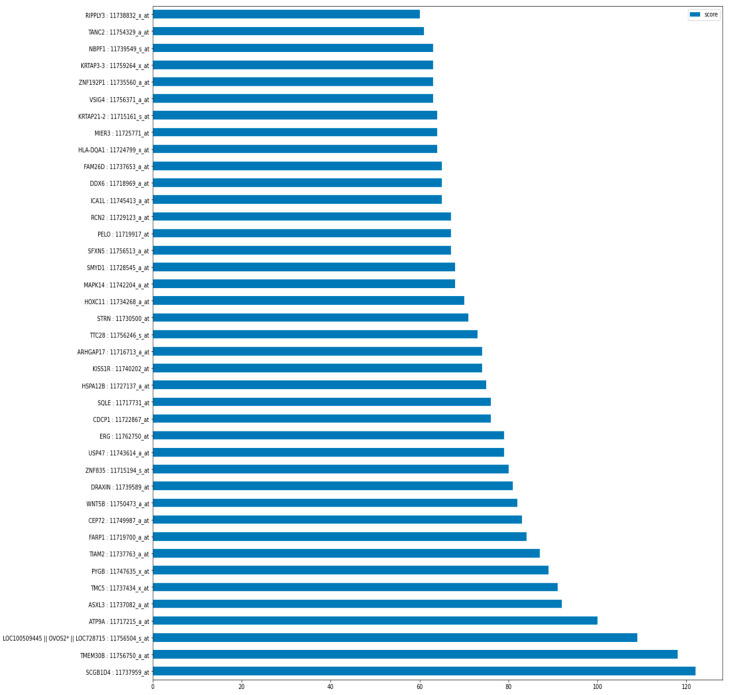
Top 40 features/gene transcripts out of 95 ranked as per feature score. The asterisk (*) in the illustration holds no specific meaning. It is part of the gene ADNI transcript name—LOC100509445 || OVOS2* || LOC728715: 11756504_S _at.

**Figure 7 diagnostics-15-00211-f007:**
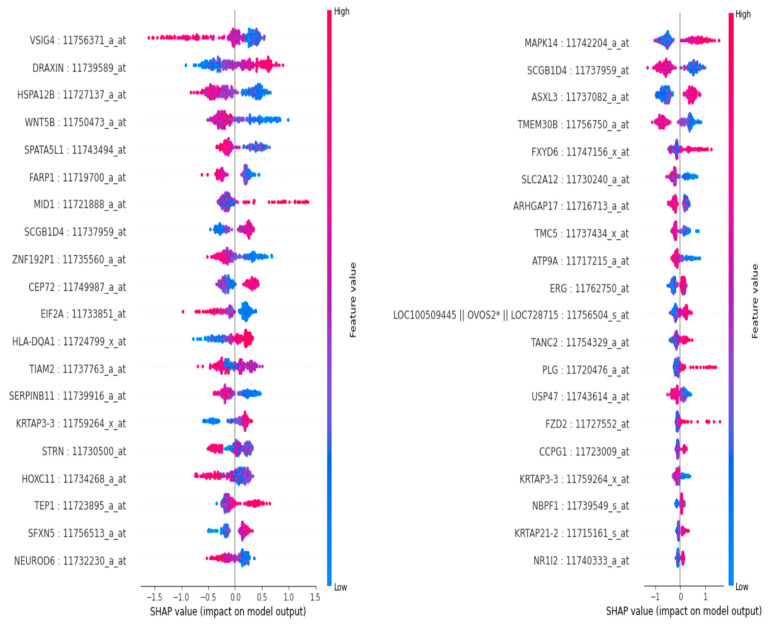
SHAP plot showing impacts of gene transcripts in identification of stages. Left to right, first—AD/dementia stage identification output from ADNI samples; second—MCI classification output from ADNI samples. The asterisk (*) in the first illustration holds no specific meaning. It is part of the gene ADNI transcript name—LOC100509445 || OVOS2* || LOC728715: 11756504_S _at.

**Figure 8 diagnostics-15-00211-f008:**
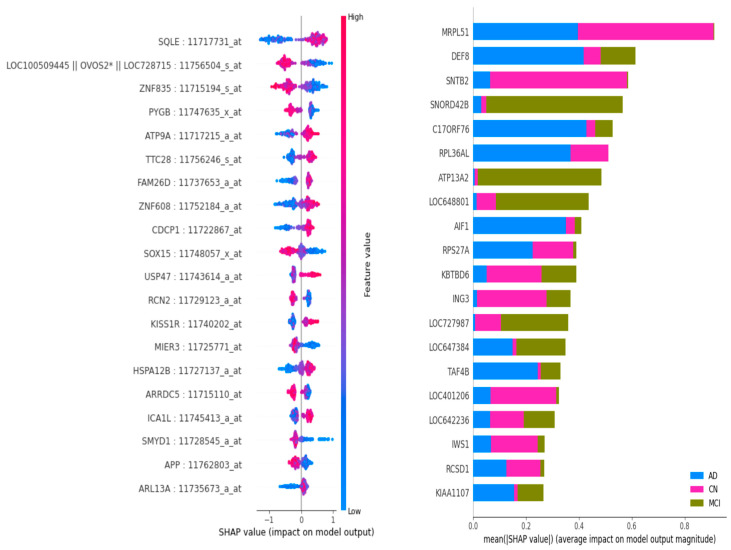
Left to right, first—SHAP plot showing impacts of gene transcripts to produce CN classification output; second—SHAP summary plot showing impacts of gene transcripts in identifying CN, AD, and MCI stages from ADNI samples. The asterisk (*) in the first illustration holds no specific meaning. It is part of the gene ADNI transcript name—LOC100509445 || OVOS2* || LOC728715: 11756504_S _at.

**Figure 9 diagnostics-15-00211-f009:**
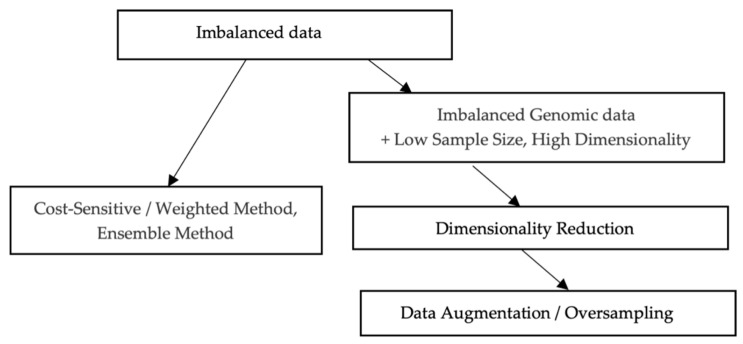
Handling of genome expression data for model construction.

**Figure 10 diagnostics-15-00211-f010:**
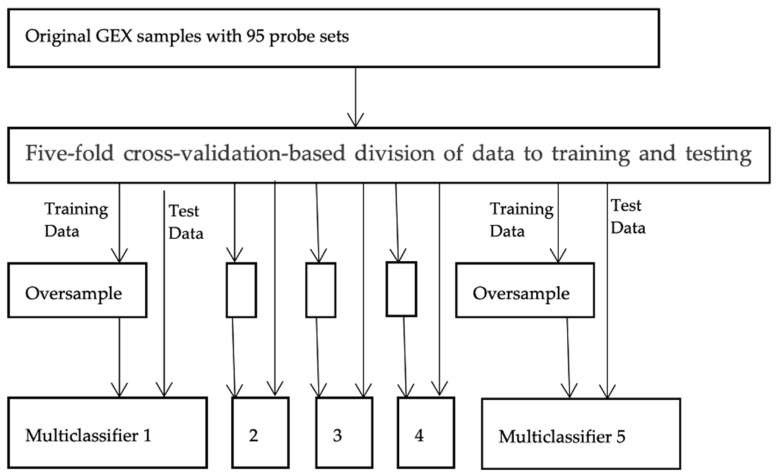
Model construction from gene expression profiles with 5-fold cross validation.

**Figure 11 diagnostics-15-00211-f011:**
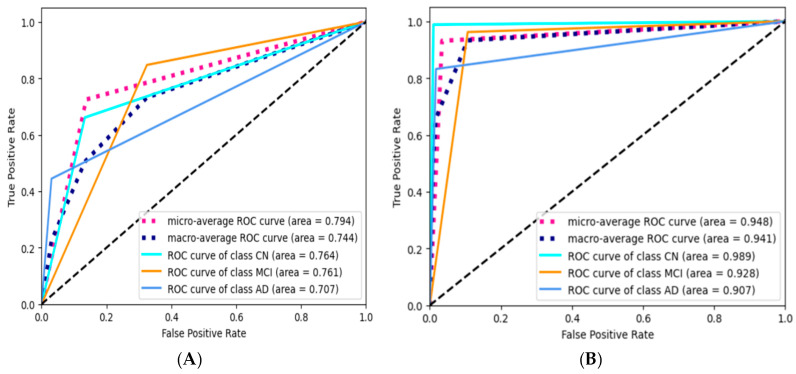
Performances of gene expression-based model and clinical data-based model. (**A**) ROC curve, gene expression-based model; (**B**) ROC curve, clinical data-based model. (**C**) PR curve, gene expression-based model; (**D**) PR curve, clinical data-based model. (**E**) CM, gene expression-based model. (**F**) CM, clinical data-based model.

**Figure 12 diagnostics-15-00211-f012:**
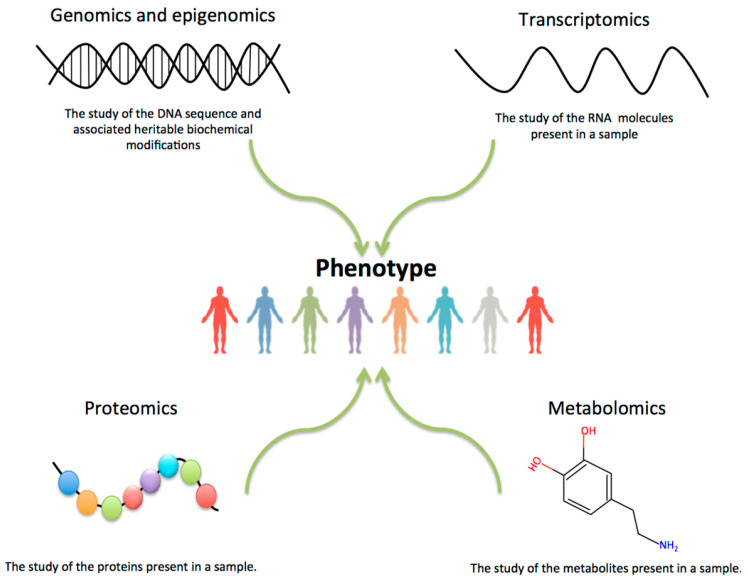
Genomic, proteomic, metabolomic, and epigenomic factors in phenotype and health. Source: EMBL-EBI, shared under a Creative Commons Attribution 4.0 International (CC BY 4.0) license.

**Figure 13 diagnostics-15-00211-f013:**
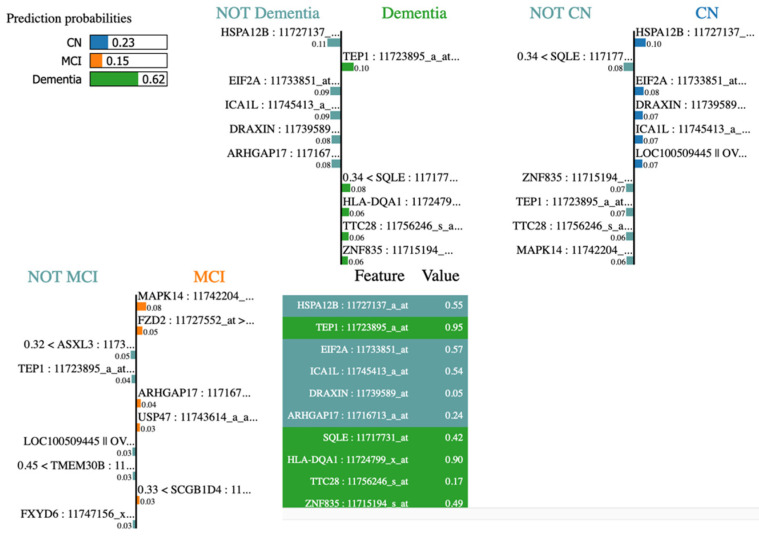
LIME plot showing contributing gene expressions leading to dementia.

**Table 1 diagnostics-15-00211-t001:** AD literature review highlights.

Study	Biomarker Dataset(s)	Feature Selection	Classifier	Results
Westman et al. (2012) [14]	Clinical CSF and baseline MRI data combination	Did not use	Orthogonal partial least squares (OPLS) tool	81.6% for CSF and 87.0% for MRI separately 91.8% accuracy for CSF and MRI together
Fulton et al. (2019) [15]	ADNI clinical study data and MRI images	GBM-based feature importance	GBM, ResNet-50	91.3% prediction accuracy with GBM, 98.99% for images with ResNet-50
Lee et al. (2020) [9]	Gene expression: ANM1, ANM2, ADNI	VAE and TF genes	LR, L1-LR, SVM, RF, and DNN	AUC: 0.657, 0.874, and 0.804 for ADNI, ANMI, and ANM2
Park et al. (2020) [16]	Gene expression: GSE33000, GSE44770, methylation data: GSE80970	Integrating DEGs and DMPs by inter-section	DNN (deep neural network)	0.823 is the average accuracy
Kalkan et al. (2022) [17]	Gene expression: GSE63060, GSE63061, and GSE140829	LASSO regression	CNN on transformed image representation	AUC of 0.875 for AD vs. CTL; AUC of 0.664 for MCI vs. AD; AUC of 0.619 for MCI vs. CTL
AlMansoori et al. (2024) [18]	ADNI gene expression, SNP, clinical data	Mutual Information (MI) feature selection, Chi-square, Lasso SHAP	SVM, Adaboost, Random Forest (RF), Multilayer Perceptron (MLP)	AUC value 0.65 for CN vs. MCI/AD diagnosis from gene expression data; 0.94 for combined SNP, clinical, and gene expression
Fathi et al. (2024) [19]	MRI images from ADNI	Resizing, removing non-brain image slices	CNN with majority-voting and probability-based ensemble methods	Accuracy of 98.57, 96.37, 94.22, 99.83, 93.88, and 93.92 for NC/AD, NC/EMCI, EMCI/LMCI, LMCI/AD, 4-way, and 3-way classification, respectively

**Table 2 diagnostics-15-00211-t002:** Gender, race, and age-wise distribution of AD, CN, and dementia of ADNI blood gene expression dataset.

Diagnosis Class	Class Sample Size	Gender		Race				Age	
		Male	Female	White	Black	Asian	Others	Age < 65	Age ≥ 65
CN	246	117	129	226	16	2	2	7	239
MCI	382	216	166	356	11	5	10	66	316
Dementia	116	75	43	108	3	4	1	6	110

**Table 3 diagnostics-15-00211-t003:** The eight selected ADNI clinical study biomarkers.

Feature Name	Feature Description
APOE4	Number of ε4 alleles
PTMARRY	Marital status
FDG	Cell metabolism measurement, reduced for AD patients
Hippocampus	Hippocampus measurement
WholeBrain	Whole brain measurement
mPACCdigit	Modified Preclinical Alzheimer Cognitive Composite with Digit
LDELTOTAL	Logical memory delayed recall total
CDRSB	Clinical Dementia Rating Scale—Sum of Boxes

**Table 4 diagnostics-15-00211-t004:** AD risk and suppressor gene list found from ADNI dataset.

Gene	Gene Nature
*MAPK14*	AD risk
*TEP1*	AD risk
*VSIG4*	AD suppressor
*ATP9A*	AD suppressor
*USP47*	AD suppressor
*KISS1R*	AD suppressor
*PLG*	AD risk
*FZD2*	AD risk
*FXYD6*	AD risk
*MID1*	AD risk
*DRAXIN*	AD risk
*WNT5B*	AD risk

**Table 5 diagnostics-15-00211-t005:** Hyperparameters for deep learning models for clinical data analysis.

Hyper Tuning Parameter	Parameter Value
Optimizer	Adam optimizer
Cost or loss function	categorical cross_entropy
Learning rate	0.001
Batch size	5
Epochs	4000
No. of layers	2
Activation function—layer 1	RELU
Activation function—layer 2	Softmax
Dropout rate	0.20

**Table 6 diagnostics-15-00211-t006:** AUC-based performance of DL, SVM, GBM, and RF models from gene expression data.

ROC Score of SMOTE-Based Models											
DL			SVM			GBM			RF		
CN	MCI	Dementia	CN	MCI	Dementia	CN	MCI	Dementia	CN	MCI	Dementia
0.717	0.681	0.634	0.672	0.651	0.597	0.711	0.667	0.548	0.593	0.575	0.620
0.763	0.761	0.706	0.690	0.648	0.660	0.659	0.660	0.574	0.617	0.597	0.534
0.736	0.644	0.602	0.761	0.664	0.572	0.678	0.635	0.577	0.613	0.578	0.527
0.669	0.668	0.607	0.694	0.647	0.586	0.678	0.655	0.639	0.639	0.592	0.619
0.732	0.729	0.613	0.668	0.641	0.572	0.663	0.655	0.618	0.598	0.574	0.601
**ROC Score of Non-SMOTE-Based Models**											
**DL**			**SVM**			**GBM**			**RF**		
**CN**	**MCI**	**Dementia**	**CN**	**MCI**	**Dementia**	**CN**	**MCI**	**Dementia**	**CN**	**MCI**	**Dementia**
0.717	0.713	0.620	0.654	0.616	0.500	0.717	0.654	0.518	0.655	0.632	0.500
0.682	0.662	0.572	0.676	0.632	0.500	0.614	0.659	0.592	0.579	0.570	0.496
0.773	0.703	0.593	0.663	0.629	0.500	0.634	0.670	0.613	0.622	0.612	0.500
0.735	0.706	0.607	0.637	0.601	0.500	0.668	0.638	0.584	0.622	0.573	0.500
0.698	0.689	0.600	0.658	0.628	0.500	0.663	0.614	0.563	0.554	0.547	0.520

**Table 7 diagnostics-15-00211-t007:** F1-score-based performance of DL, SVM, GBM, and RF models from gene expression data.

F1-Score of SMOTE-Based Models											
DL			SVM			GBM			RF		
CN	MCI	Dementia	CN	MCI	Dementia	CN	MCI	Dementia	CN	MCI	Dementia
0.63	0.71	0.39	0.57	0.70	0.32	0.62	0.70	0.20	0.45	0.64	0.37
0.71	0.77	0.53	0.60	0.67	0.41	0.55	0.68	0.25	0.46	0.65	0.18
0.63	0.69	0.33	0.67	0.71	0.27	0.55	0.67	0.29	0.44	0.66	0.17
0.52	0.70	0.37	0.56	0.69	0.36	0.53	0.68	0.43	0.48	0.65	0.39
0.65	0.73	0.36	0.55	0.67	0.28	0.53	0.69	0.36	0.42	0.64	0.33
**F1-Score of Non-SMOTE-Based Models**											
**DL**			**SVM**			**GBM**			**RF**		
**CN**	**MCI**	**Dementia**	**CN**	**MCI**	**Dementia**	**CN**	**MCI**	**Dementia**	**CN**	**MCI**	**Dementia**
0.63	0.75	0.37	0.52	0.71	0.00	0.62	0.72	0.09	0.51	0.74	0.00
0.58	0.69	0.25	0.57	0.69	0.00	0.45	0.71	0.30	0.32	0.68	0.00
0.68	0.73	0.32	0.51	0.74	0.00	0.48	0.75	0.36	0.42	0.74	0.00
0.62	0.75	0.36	0.47	0.70	0.00	0.52	0.70	0.30	0.43	0.69	0.00
0.60	0.72	0.33	0.52	0.71	0.00	0.53	0.68	0.24	0.27	0.67	0.08

**Table 8 diagnostics-15-00211-t008:** Performance of the DL model constructed using clinical data.

F1 Score			ROC AUC		
CN	MCI	Dementia	CN	MCI	Dementia
0.983	0.923	0.847	0.983	0.937	0.877
0.971	0.939	0.886	0.989	0.927	0.907
0.944	0.931	0.869	0.967	0.921	0.915
0.968	0.933	0.846	0.968	0.944	0.897
0.947	0.856	0.823	0.941	0.893	0.908

## Data Availability

All data obtained by analyzing this research are from the authors and are available within the manuscript. Data used in this research are accessible from the Alzheimer’s Disease Neuroimaging (ADNI) database on the LONI website (adni.loni.usc.edu).

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
