# Peer review of "Machine Learning-Based Alzheimer’s Disease Stage Diagnosis Utilizing Blood Gene Expression and Clinical Data: A Comparative Investigation"

_diagnostics, 2025, doi:10.3390/diagnostics15020211_

Round 1

Reviewer 1 Report (Previous Reviewer 2)

Comments and Suggestions for Authors

The requested revisions have been made by the author. The article is seen to be highly cited. Therefore, I accept. The figures can be made more visible.

Author Response

Reviewer 2 Report (New Reviewer)

Comments and Suggestions for Authors

In the study, authors have presented a comparative analysis of multi-stage diagnosis of Alzheimer's disease (AD), including Mild Cognitive Impairment (MCI), utilizing two distinct types of biomarkers: blood gene expression and clinical biomarker samples. The results are promising and will be attractive to the related fields. However, there are some minor issues that authors need to solve before publishing. 

1. Could authors explain why they used multiple classifiers (DL, SVM, GB, RF) for gene expression data but only DL for clinical data?

2. Given the class imbalance, did authors consider using other metrics besides ROC AUC and F1 scores (e.g., precision-recall curves, balanced accuracy)?

3. How stable were the authors' results across different random seeds and cross-validation folds?

4. Could the authors also highlight the total sample size for each class (CN, MCI, AD) in both the clinical and genetic datasets?

Author Response

This manuscript is a resubmission of an earlier submission. The following is a list of the peer review reports and author responses from that submission.

Round 1

Reviewer 1 Report

Comments and Suggestions for Authors

-The overall structure of the paper is satisfactory.

-The manuscript does not provide a well-defined background, making it difficult to understand the context of the study.

-The study's objectives are not explicitly stated, making it hard to discern the aim and purpose of the research.

-The manuscript lacks innovative contributions and does not present new findings or methods in the field.

-The methodology section is not clearly written, leaving readers uncertain about the approach taken and how the study was conducted.

-The paper does not offer any new insights or findings compared to existing state-of-the-art approaches in the domain.

-There is no comparison made between the proposed method and existing methods in the literature, which would help in evaluating the effectiveness of the approach.

-The authors have cited older references, missing recent developments and up-to-date studies in the field.

-The figures in the manuscript lack clarity, making it difficult for readers to interpret the presented information.

-Why datasets are treated as separate concepts, and no reasoning or inference techniques are applied. For instance, characteristics such as feathers, wings, and the ability to fly could be used to classify an object as a bird. Why the dataset is not designed as per above mentioned example and etc.

-The manuscript does not explain how 'Gene nature'  information in Table 4 was considered, leaving out a crucial detail in the analysis.

Reviewer 2 Report

Comments and Suggestions for Authors

The introduction section of the article summarizes the basic information about Alzheimer's disease (AD) well. However, the originality of this study should be emphasized more clearly. For example, the statement "this study is the only study that diagnoses the stages of Alzheimer's" should be compared with similar studies that have been done before. In addition, some important studies from the latest literature should be added, so a more extensive literature review would be useful.

It is seen that cross-validation was not performed due to the small data set used in the study. Not performing cross-validation may increase the sensitivity of the model to overfitting. Cross-validation techniques suitable for the dimensions of the data set, such as k-fold, may be preferred.

Using XGBoost and SFBS algorithms in feature selection is a positive approach.

The deep learning model used for clinical data is a suitable choice since there is sufficient data. However, for data with high dimensions and low sample numbers, such as gene expression, comparative results of traditional machine learning methods such as SVM or Random Forest can also be used instead of deep learning. If machine learning methods are not to be used, it should be explained with a strong reference why deep learning methods are preferred.

The authors have correctly used measures such as F1-score and ROC AUC due to data imbalance. The source should be left on how the performance values ​​(Accuracy, F1-score, recall, roc, etc.) were obtained. You can provide a source for this. For example: https://www.sciencedirect.com/science/article/pii/S0208521622000432  (Section 4. Experimental analysis and results)

In addition, the reasons why the F1-score is at a low level of 0.67, especially for gene expression data, should be explained in more detail.

It is not clear with the Confison matrix. The texts containing numerical values ​​should be more visible.

Comments on the Quality of English Language

Language is normal, no problem.
